# Identification of Quantitative Trait Locus and Candidate Genes for Drought Tolerance in a Soybean Recombinant Inbred Line Population

**DOI:** 10.3390/ijms231810828

**Published:** 2022-09-16

**Authors:** Wenqi Ouyang, Limiao Chen, Junkui Ma, Xiaorong Liu, Haifeng Chen, Hongli Yang, Wei Guo, Zhihui Shan, Zhonglu Yang, Shuilian Chen, Yong Zhan, Hengbin Zhang, Dong Cao, Xinan Zhou

**Affiliations:** 1Key Laboratory of Biology and Genetic Improvement of Oil Crops, Ministry of Agriculture and Rural Affairs, Oil Crops Research Institute of Chinese Academy of Agricultural Sciences, Wuhan 430062, China; 2The Industrial Crop Institute, Shanxi Academy of Agricultural Sciences, Taiyuan 030006, China; 3Crop Research Institute, Xinjiang Academy of Agricultural and Reclamation Science, Key Laboratory of Cereal Quality Research and Genetic Improvement, Xinjiang Production and Construction Crops, Shihezi 832000, China

**Keywords:** soybean, drought, RIL, resequencing, QTLs

## Abstract

With global warming and regional decreases in precipitation, drought has become a problem worldwide. As the number of arid regions in the world is increasing, drought has become a major factor leading to significant crop yield reductions and food crises. Soybean is a crop that is relatively sensitive to drought. It is also a crop that requires more water during growth and development. The aim of this study was to identify the quantitative trait locus (QTL) that affects drought tolerance in soybean by using a recombinant inbred line (RIL) population from a cross between the drought-tolerant cultivar ‘Jindou21’ and the drought-sensitive cultivar ‘Zhongdou33’. Nine agronomic and physiological traits were identified under drought and well-watered conditions. Genetic maps were constructed with 923,420 polymorphic single nucleotide polymorphism (SNP) markers distributed on 20 chromosomes at an average genetic distance of 0.57 centimorgan (cM) between markers. A total of five QTLs with a logarithm of odds (LOD) value of 4.035–8.681 were identified on five chromosomes. Under well-watered conditions and drought-stress conditions, one QTL related to the main stem node number was located on chromosome 16, accounting for 17.177% of the phenotypic variation. Nine candidate genes for drought resistance were screened from this QTL, namely *Glyma.16G036700*, *Glyma.16G036400*, *Glyma.16G036600*, *Glyma.16G036800*, *Glyma.13G312700*, *Glyma.13G312800*, *Glyma.16G042900*, *Glyma.16G043200,* and *Glyma.15G100700*. These genes were annotated as NAC transport factor, GATA transport factor, and BTB/POZ-MATH proteins. This result can be used for molecular marker-assisted selection and provide a reference for breeding for drought tolerance in soybean.

## 1. Introduction

Drought stress is one of the major environmental factors that cause changes in phenotypic, physiological, biochemical, and molecular levels in plants [1,2]. These changes adversely affect plant growth, plant development, and crop production. In total, 65% of global freshwater use is devoted to the growth stage of the plant. Severe drought would cause the termination of photosynthesis and disruption of the metabolism, and finally, lead to plant death [3,4]. The study of drought resistance in crops is very important. The identification of drought resistance genes and the exploration of drought-resistance mechanisms in plants is of vital importance as a means of breeding new varieties of drought-tolerant crops.

Soybean (*Glycine max*) is an important legume crop that can be processed into a variety of soybean oils or used to feed livestock and is a globally important cash and food crop [5]. As a crop with high water requirements [6], soybean is extremely sensitive to water deficits. Soybean requires irrigation of 1300–2200 g of water for every 1 g of seed formed at maturity [7], making it one of the most sensitive legume crops to water deficit [8]. Continued global drought has caused an approximately 40% loss in quality and yield of soybean [9]. Improving the drought resistance of soybean and selecting new drought-resistant soybean varieties is an important way to ensure high and stable yields in soybean production [10].

Conventional breeding enables the recombination of excellent drought-tolerant genes in high generations of crops [11,12], which is costly and time-consuming. The emergence of molecular marker-assisted breeding has become an important tool for studying genomic diversity and identifying domesticated selective regions, the key quantitative trait locus (QTL), and genes for important traits [13,14,15]. The QTL associated with grain yield under drought stress was identified using SSR markers in *Oryza sativa* L. [16]. With the rapid development of high-throughput sequencing technologies, QTL mapping and genome-wide association analysis (GWAS) have been widely used for the genetic analysis of drought tolerance traits in crops [17,18,19], including rice (*Oryza sativa* L.) [20,21], wheat (*Triticum aestivum* L.) [22,23], maize (*Zea mays* L.) [24], and soybean (*Glycine max*) [25]. The discovery of drought-resistance QTL and the screening of candidate drought-resistance genes based on agronomic traits associated with drought resistance in crops has greatly contributed to the development of drought-resistant crop breeding.

Drought tolerance is a complex quantitative trait. Many traits have changed under drought stress. Previous studies have indicated that some physiological traits, such as relative water content, relative electrical conductivity, and chlorophyll content, and some yield-related traits, such as plant height, number of nodes in main stems, leaf area, pod number, seed number, and seed weight, are affected by soil drought stress, in addition to the significant reduction in total dry matter and yield of soybean plants during the growth and flowering stages [26]. These traits can be considered indicators to judge the drought tolerance of the crop [27,28,29]. It is therefore necessary to have an evaluation of soybean phenotype, physiological, and yield-related traits for drought tolerance.

Some drought-tolerant phenotypic and physiologically related QTL have been identified in soybean, such as relative water content, the relative electrical conductivity of soybean leaves, and water use efficiency [23,27,30]. Drought is a complex multi-gene controlled quantitative trait [24], which influences plant height, plant weight, node number, and yield [31,32,33,34,35]. Hwang et al. [34] constructed five recombinant inbred line (RIL) populations to analyze loci controlling leaf wilting in soybean, and a total of seven stable QTLs were localized in the populations. A total of 136 soybean drought-tolerant lines were tested for SNPs between drought-tolerant and sensitive genotypes, and 13 genes associated with the number of nodes in main stems were identified [33]. By GWAS, Zhang et al. identified 53 QTLs in 19 soybean chromosomes, with two single nucleotide polymorphisms (SNPs) associated with plant height falling within the confidence interval of two QTLs in different water conditions [32]. In 373 soybean varieties, 31,260 SNPs were obtained, and 47 SNP loci associated with water use were located by GWAS [36]. The use of molecular markers to assist in breeding selection for identifying drought-tolerant QTL in soybean can greatly improve the efficiency of selection for drought-tolerant varieties of soybean [36].

In the present study, we constructed a high-density map by whole genome resequencing techniques (WGRS) of 162 soybean RIL lines generated by drought-tolerant cultivar Jindou21 and resequencing drought-sensitive cultivar Zhongdou33. Some agronomic traits (node number of the main stem (NNMS), chlorophyll content (CC), branches (BN), pull stem (PS), leaf area (LA), plant height (PH), biomass, seed weight per plant (SWPP), and maturity) and physiological traits (Relative water content of leave (RWCL) and Relative electric conductivity of leave (RECL)) were analyzed to identify QTLs and candidate genes for drought tolerance under well-watered conditions and drought stress conditions, by using a water-catch tank to simulate drought stress [37]. This research provides some valuable information for understanding the molecular basis and breeding for drought tolerance.

## 2. Results

### 2.1. Effect of Genotype and Water Status on Agronomic and Physiological Traits

The QQ chart was used to describe the distribution of 11 traits in soybean RILs. The results showed that the majority of traits show a normal distribution except PS and M under well-watered and drought-stress conditions (Figure 1). The drought condition was simulated by water-catch tank treatment. Field traits were measured at the beginning of the seventh week when there were the most significant differences in soil water contents between the two treatment conditions (Appendix A). Therefore, mean square and significance tests were conducted using combined ANOVA of NNMS, CC, RWCL, BN, LA, PH, BIOMASS, SWPP, RWCL, and RECL to investigate the effects on agronomic and physiological traits under different genotypes and environments (Table 1). The results indicated that CC, RECL, PH, NNMS, BN, RWCL, biomass, and SWPP were highly significantly different (*p* < 0.01) among the RILs and water treatments. The interaction of different lines and water status has a highly significant effect on CC, LA, PH, NNMS, BN, RWCL, and BIOMASS (*p* < 0.01), a significant effect on SWPP (*p* < 0.05), and no significant effect on the trait RECL.

### 2.2. Traits Correlation and Principal Component Analysis

Figure 2 describes the correlation of seven agronomic traits and two physiological traits under well-watered and drought conditions. The correlation coefficients (r) indicate the degree of correlation between these traits. NNMS had a highly positive correlation with PH (*p* < 0.001, r > 0.6); similarly, NNMS and PH had a strong positive correlation with biomass (*p* < 0.01, r > 0.3) under well-watered drought stress conditions. In addition, RECL showed a significantly negative correlation with PH and biomass, respectively. Biomass had a positive correlation with SWPP under well-watered and drought stress conditions (*p* < 0.001, r > 0.6). Thus, NNMS and PH may be the key traits affecting biomass and SWPP under well-watered and drought-stress conditions.

The relationships between genotypes and traits have been further investigated by principal component analysis of double-labeled plots under well-watered and drought stress conditions (Figure 3). The smaller the angle between the dimensional vectors of each trait in the plots, the higher the correlation between the traits. S151, S157, S161, and S152 had higher biomass that was mainly contributed by two traits NNMS and PH under drought stress conditions. S161 showed higher yields also mainly contributed by NNMS and PH.

### 2.3. Linkage Mapping and QTL Analysis

The depth of the parental genomes was above 20X. The average coverage of the genomes was above 90%. The average depth of the offspring samples was 4.21X, with a coverage of above 92.86%. There was a total of 20 chromosomes in the high-density genetic map (Figure 4), containing 923,420 SNPs with a well-distributed linkage distance across the chromosomes. A total of 18 QTL associated with drought resistance were detected in 20 chromosomes under both normal and stress environments. Of these, under well-watered conditions, nine QTLs were localized on chromosomes 05, 08, 12, 13, 15, 16, 19, and 20, and nine QTLs were localized on chromosomes 02, 08, 11, 13, 15, 16, and 18 under drought stress (Table 2).

One QTL localized on chromosome 16 was identified for NNMS under well-watered and drought conditions, which may be a major QTL. It contributed 17.18% and 15.24% of the phenotypic variation, with maximum LOD values of 8.70 and 7.51 under well-watered and drought conditions, respectively (Figure 5).

Furthermore, two QTLs located on chromosomes 16 and 20 were identified for PH under well-watered conditions, and one was located on chromosome 13 under drought stress (PVE > 8%). One QTL identified for BN was located on chromosome 19 with a LOD value of 4.2 and 7.5% of PV under the well-watered condition. Under drought conditions, a QTL associated with RWCL located on chromosome 16 was identified, explaining up to 10% of the PV, while a QTL for RECL localized on chromosome 8 explained 9.2% of the phenotypic variation. For BIOMASS, two QTLs were identified on chromosomes 5 and 15, with LOD values above 2.5, and contributing 6.6 and 10.4%, respectively, under the well-watered condition, while one QTL localized on chromosome 11 had a LOD value of 3.9 and a PVE value of 7.6% under the drought condition. For SWPP, two QTLs located on two chromosomes were identified, explaining 3.85% and 5.07% of the PV with maximum LOD values of 2.23 and 2.44, respectively.

### 2.4. Prediction of Candidate Genes

Based on the results of the locus, 135 candidate genes of the major QTL for NNMS were identified under both drought and well-watered conditions (Table 3). The results showed that one QTL was located at 13.769 cM on chromosome 16, with 135 genes in the interval. The results of Gene Ontology (GO) enrichment and classification revealed that these genes are involved in the regulation of plant growth and development and response to a stimulus (Figure 6). These candidate genes are annotated as different kinases, transcription factors, and functional proteins (Table 3). The kinases mainly include mitogen-activated protein kinase (MAPK) and adenylyl-sulfate kinase (ASK). NAC, GATA transcription factor, ethylene-responsive transcription factor, and RAP-like transcription factors are also identified. In addition, some functional proteins, such as nucleoside-triphosphatase, BTB/POZ MATH (BPM) protein, and the plant flowering control gene FLOWERING LOCUS T (FT), were also annotated. It is worth noting that several candidate drought-tolerant genes were identified, such as NAC transport factor (*Glyma.16G042900* and *Glyma.16G043200*) and GATA transport factor (*Glyma.16G042300*), and BPM proteins, such as *Glyma.16G036700*, *Glyma.16G036400*, *Glyma.16G036600*, and *Glyma.16G036800*, under drought and well-watered conditions (Table 3). These gene families were involved in the regulation of drought stress [38,39,40,41,42,43,44,45].

Under water-deficiency treatment, some candidate genes associated with drought resistance based on RECL were localized in chromosome 15 within 39.392–39.706 M, including a drought-resistance gene methionine sulfoxide reductase (MSR), *Glyma.15G100700*, which regulates chloroplasts. Four candidate genes were identified from 235 candidates with a drought stress response and PH on chromosome 13, annotated as PUB and NAC genes, respectively.

## 3. Discussion

### 3.1. The Character of the WGRS Approach to Identify QTL Makers and Candidate Region Analysis

The use of genetic maps is essential for finding important loci, precision mapping, and marker-assisted breeding [46]. Several genetic maps have been constructed for soybean based on molecular markers such as SSRs, ESTs, RFLP, and RAPD markers [47,48]. Using DNA markers, it was possible to identify regions on the genetic map that could be identified by the main genes, in accordance with standard mapping procedures. However, it is difficult to find the key locus associated with a specific trait by using non-specificity makes [49].

By advancing DNA sequencing technologies and applications, scientists have been able to improve plant breeding and aid in fine mapping processes in the past decade, as well as discover new types of molecular markers. Regarding detecting SNP markers and accurate genotyping, a high-density genetic map can be constructed with the next-generation sequencing technologies developed and the soybean reference genome sequence published [50,51]. The SNP molecular markers’ identification by WGRS has been well used for studying drought resistance in many important crops [33,34].

In this study, the parental lines and 162 RILs were sequenced by WGRS to construct a high-density genetic map. Among them, the parental lines had a 20-fold sequencing depth and average genome coverage of 90% or more, and each RIL had a 4.21-fold sequencing depth and coverage of 92.86% or more. Ultimately, 32.84 Gbp of high-quality reads from Jindou 21, 33.3 Gbp of high-quality reads from Zhongdou33m and 748.99 Gbp of high-quality reads from their progeny were obtained. By utilizing bin markers and accurate genotypic data, it was possible to construct a high-density genetic map. An analysis of genotyping data showed that 4843 recombination bin markers represented 923,420 SNPs on 20 linkage groups. There was an average distance of 0.57 cM between adjacent bin markers on the linkage map, with a total length of 2,737.51 cM. The collinearity of the genetic maps and physical maps was good for each linkage group (Appendix A). Here, we demonstrate that the WGRS strategy is an effective tool for detecting markers and building high-density linkage maps. The WGRS mapping enabled us to obtain a great number of genome-wide SNPs, which accurately reflect the genetic diversity and genomic diversity of soybean.

### 3.2. Yield-Related Traits Analysis

There have been many studies on the identification of drought tolerance in soybean varieties and the screening of drought-resistant germplasm resources, in which the identification of drought-tolerance indicators and traits in soybean varieties is a crucial step. In previous studies, drought-tolerance indicators mainly include yield traits, growth and development indicators, morphological indicators, and physiological and biochemical indicators [10,52]. Soybean drought-tolerance traits include leaf wilting [53,54], root morphology [53], yield under drought [54], etc. The drought treatment of soybean plants has also been shown to cause changes in the number of branches and main stem nodes and plant height [32,33].

In legumes, especially soybean, there are many studies on NNMS. NNMS is an important trait for soybean breeding. Soybean canopy and seed yield are determined by NNMS, which is one of the major plant agronomic traits [55]. Furthermore, there was a correlation between NNMS and other important agronomic traits, such as plant height, flowering, and maturity [56]. Li et al. reported that some QTLs associated with NNMS for plant density were identified using 144 four-way recombinant inbred lines (FW-RILs). The candidate genes were found on chr 06 and chr 19, named *Glyma.06G094400, Glyma.06G147600*, *Glyma.19G160800*, and *Glyma.19G161100* [57]. Fu et al. utilized 306 accessions from northeast China to identify 76 QTLs associated with NNMS for yield and identified 49 candidate genes [58]. Plant height is one of the main hot spots in plant abiotic stress. There are many studies on the association between plant height and drought tolerance. In soybean, six QTLs for drought tolerance associated with plant height have been identified (qPH2, qPH6, qPH7, qPH19-1, qPH19-2, and qPH19-3) [59,60].

In our study, there is a strong relationship between NNMS and PH (*p* < 0.001) under well-watered and drought-stress conditions. PH was also highly correlated with BIOMASS and Y traits under both conditions. The above result is consistent with earlier reports. Moreover, the correlation between NNMS and BIOMASS under the drought stress condition was enhanced compared to the well-watered condition. It is tempting to speculate that NNMS and PH are traits associated with drought tolerance.

### 3.3. Preliminary Analysis of the Potential Functions of Candidate Genes

In this study, the main QTL was localized with NNMS under both drought and well-watered conditions, and the candidate genes in this region were annotated with GATA, NAC transcription factors, and BPM proteins in Arabidopsis. In addition, under drought conditions, several candidate genes associated with RECL and PH were identified including MSR, PUB, and NAC genes.

Protein degradation is essential for plant growth and development. The BPM protein is part of the Cullin E3 ubiquitin ligase complex [39,61], and binds at least three transcription factor families—ERF/AP2 class I, the homologous cassette-leucine zip, and R2R3 MYB—to degrade target proteins by ubiquitination, which plays an important role in plant abiotic stress responses, especially drought resistance [39]. In this study, the main QTLs associated with NNMS have been identified. Four candidate genes, including *Glyma.16G036700*, *Glyma.16G036400*, *Glyma.16G036600,* and *Glyma.16G036800*, were annotated as BPM4-like proteins. Their homologous gene *AT3G03740* (BPM4) has been reported to interact with the transcription factor ERF/AP2 to regulate drought tolerance in Arabidopsis [62]. The plant U-box (PUB) gene family is also a major family of ubiquitin ligases in plants. They are involved in protein degradation pathways and physiological processes regulated by drought stress in plants [63,64]. For example, in Arabidopsis, PUB22 and PUB23 coordinate the regulation of drought signaling pathways through the ubiquitination of cytoplasmic RPN12a to enable plants to respond to water deficits [65]. In addition, the expression of PUB6 in soybean leaves and roots is induced by abscisic acid (ABA), high salinity, and polyethylene glycol (PEG), which play a negative regulatory role in drought tolerance [66]. Here, two candidate genes, *Glyma.13G312700* and *Glyma.13G312800,* associated with PH, were identified. Their homologs gene *PUB23*, encoding E3 ubiquitin protein ligase, negatively regulates drought tolerance by controlling the ABA receptor PYL9 in Arabidopsis [67].

The NAC (NAM, ATAF1/2, CUC2) protein family is a plant-specific transcription factor superfamily in most plants [40,41,42,68]. Wu et al. [42] revealed that PtrNAC72, a blocker of putrescine biosynthesis, may negatively regulate plant response to drought stress by acting as a deterrent to putrescine-related ROS homeostasis. Under dehydration stress conditions, ANAC096 cooperates with the bZIP-type transcription factors ABRE binding factor and ABRE binding protein (ABF/AREB) to support plants’ survival [43]. In soybean, the overexpression of NAC085 decreases malondialdehyde content and increases superoxide dismutase, catalase, and ascorbate peroxidase activities under abiotic stresses [44]. In the study, *Glyma.16G042900* and *Glyma.16G043200* associated with NNMS were annotated as NAC100-like and NAC18-like, respectively. Their homologs genes were NAC87 (*AT5G18270*) and NAC18 (*AT3G04070*). In Arabidopsis, ding et al. identified dehydration stress memory response genes based on genome-wide RNA-Seq, in which NAC87 (*AT5G18270*) and NAC18 (*AT3G04070*) were associated [45].

MSR genes play an important role in plant stress resistance. MSR catalyzes the reduction of methionine sulfoxide to methionine residues. Reactive oxygen species (ROS) caused by biotic and abiotic stresses in plants lead to protein denaturation. MSR proteins can reduce plant damage during ROS disruption. For example, the pepper MSR2 is responsible for reducing oxidized porphobilinogen deaminase (PBGD), which can protect chlorophyll synthesis under drought conditions [38]. The MSR2 gene (*Glyma.15G100700*), which was screened by a QTL for RECL, could play an important role in drought tolerance in soybean. These candidate genes will require subsequent validation experiments to elucidate their drought-tolerance functions.

## 4. Materials and Methods

### 4.1. Plant Material and Growing Conditions

A RIL population developed from a cross between a drought-tolerant cultivar ‘Jindou21’ and the drought-sensitive cultivar ‘Zhongdou33’. The parents and 160 RILs of F8 were grown in the field of the Industrial Crop Institute, Shanxi Academy of Agricultural Sciences, China in 2019.

### 4.2. Experimental Design and Drought Conditions

The accurate identification of drought tolerance in the field is difficult. The effect and feasibility of the collecting trough have been evaluated. Placing the collecting trough between soybean rows could reduce soil water content [37]. Here, the collecting troughs with a diameter of 25 cm were placed in the field in the vegetative period to collect rain. The method was used to simulate drought treatment to identify phenotypic and agronomic traits of the RIL lines. A soil moisture meter (FIELDSCOUT TDR 100, Campbell) was used to measure the soil water content under the control and drought stress conditions.

### 4.3. Measurement of Traits and Phenotyping

Drought-tolerance indicators were identified in the field and the greenhouse in 2019. The plant height, branch number, chlorophyll content, relative water content of leaves, the relative electric conductivity of leaves, the node number of the main stem, pull stem, and leaf area were measured at the R2 stage, whereas the biomass (including seeds), seed weight per plant, and maturity were measured when the plant was harvested at the R8 stage. Leaf area was determined using a portable leaf area meter (YMJ-D). Chlorophyll content was determined using a portable chlorophyll meter SPAD502. Stem strength was determined using a SUNDOO portable tester. Leaf relative water content and electrical conductivity were determined using the method in [69,70,71]. Each drought coefficient (DC) of 11 traits was calculated as the ratio of the individual trait under drought stress condition or well-watered conditions as shown in the equation below.
DC = Trait in Drought Stress Condition/Trait in Well-watered Condition

Five plants were measured per replicate of each trait (n = 3 biological replicates).

### 4.4. DNA Extraction, DNA Sequencing and SNP Identification

A total of 162 RILs, Jindou21, and Zhongdou33 genomic DNA were extracted from fresh young leaves using the Plant DNA Kit (D2485, Omega). In order to sequence each DNA sample, paired-end sequencing libraries were constructed with an insertion of 300–500 bp. An Illumina Hiseq 2000 system (Illumina Inc., San Diego, CA, USA) was used to sequence the libraries with a 150 bp (PE150) read length. Low-quality reads (quality score < 20e) were filtered out, and then raw reads were sorted to each sample according to barcode sequences. We used Samtools v 1.9 [72,73] to mark duplicates and GATK v 2.8.1 [74] to realign local elements and calibrate bases. With the default parameter settings, GATK and Samtools [73], SNP calling analysis was formed to produce a set of SNPs. There are eight segregation patterns based on polymorphic SNPs between parents (ab × cd, ef × eg, hk × hk, lm × ll, nn × np, aa × bb, ab × cc, and cc × ab). Only SNPs with the aa×bb pattern were chosen for further analysis.

### 4.5. Construction of Linkage Map

A total of 32.84 Gbp of clean data was obtained for parent Jindou21 and 33.3 Gbp for Zhongdou33. In total, 160 offspring had a total of 748.99 Gbp of data. Q30 in 162 RILs reached over 80%. A total of 1,913,252 SNPs were detected between the parents and were screened for redundancy.

To determine recombination breakpoints and construct a bin map of RILs, a slightly modified sliding window approach was employed [75]. The ratio of SNPs with ‘Jindou21′ and ‘Zhongdou33′ genotypes was calculated. According to Huang et al. [75], a physical bin map was constructed based on the recombination breakpoint position. Briefly, when RILs did not have recombination breakpoints within a 100 kb interval, these regions were combined into one bin. When recombination breakpoints did not occur within 100 kb intervals, the regions were combined into a single bin.

A genetic map with ultra-high density was constructed using bins as markers. Molecular markers were divided into linkage groups (LGs) based on their locations on the genome. To further confirm the robustness of markers in each LG, modified logarithms of odds (MLODs) were calculated between markers. Before ordering, markers with MLOD scores < 5 were filtered. For resolving genotyping errors within LGs, the HighMap [76] strategy was utilized. The SMOOTH error correction strategy is then applied based on parents’ genotype contributions, and a k-nearest neighbor algorithm is used to impute missing genotypes. After applying the multipoint method of maximum likelihood, skewed markers were added to the map. Kosambi mapping function [77] was used to estimate map distances.

### 4.6. QTL Analysis

The effects of genotype and environment, as well as interaction effects between genotype and environment, were estimated using an analysis of variances (ANOVA). It was estimated that broad sense heritability is derived from the formula H^2^ = σ^2^_g_/(σ^2^_g_ + σ^2^_ge_/n + σ^2^_e_/nr), where n delegates the number of environments, r delegates the number of replications, σ^2^_g_ delegates the estimated genetic variance, σ^2^_ge_ delegates variance for genotype-environment interaction, and σ^2^_e_ delegates the experimental error. R package R/qtl was used to identify QTLs via composite interval mapping (CIM) [78]. LOD values were determined based on a 1000-permutation test. QTLs were called for LOD values of 2 and higher. A 1000-permutations test was used to determine LOD values. When LOD values exceeded 2, QTLs were identified.

### 4.7. Analysis of Phenotypic Data

The ANOVA analysis was performed using the SPSS 19 (George 2012) following a chi-squared test. To describe the magnitude of the relationships between physiological and agronomic traits, Pearson correlation coefficients (r) were calculated separately for the stress and non-stress treatments using R version 4.0.1 (2 July 2021). Correlation coefficients were calculated using the Pearson model. Normal distribution tests were plotted using the R package qqman, phenotypic correlation association plots were plotted using the R package Performance Analytics, and principal component analyses were plotted using the R packages FactoMineR, factoextra, and corrplot.

### 4.8. Potential Candidate Genes Prediction

The genes within QTLs were searched for potential candidates. The candidate genes were identified as those linked directly to drought stress or those that were associated with stress. A search for drought stress-related gene names and functions was carried out in Soybase (www.soybase.org, 13 June 2022), NCBI (https://www.ncbi.nlm.nih.gov/, 13 June 2022), and Phytozyme (https://phytozome.jgi.doe.gov, 13 June 2022).

## 5. Conclusions

A genetic map of 20 linkage groups with a total distance of 2737.51 cM was developed using 162 soybean recombinant inbred line populations and 923,420 SNP markers. The genetic map was used to identify drought-tolerant QTLs for traits in soybean. In this study, the main QTL on chromosome 16 was identified for NNMS under well-watered and drought conditions, which explained more than 10% phenotypic variation and had a LOD score larger than 6. Several of the candidate genes in this region were associated with NAC, BPM, and PUB proteins, which possibly enable plants to respond to drought stress, namely *Glyma.16G036700*, *Glyma.16G036400*, *Glyma.16G036600*, *Glyma.16G036800*, *Glyma.13G312700*, *Glyma.13G312800*, *Glyma.16G042900*, *Glyma.16G043200,* and *Glyma.15G100700*. The QTLs and candidate genes detected in this study could provide an important step toward clarifying the mechanisms of drought tolerance in soybean and further provide a theoretical basis for drought-tolerant breeding in soybean.

## Figures and Tables

**Figure 1 ijms-23-10828-f001:**
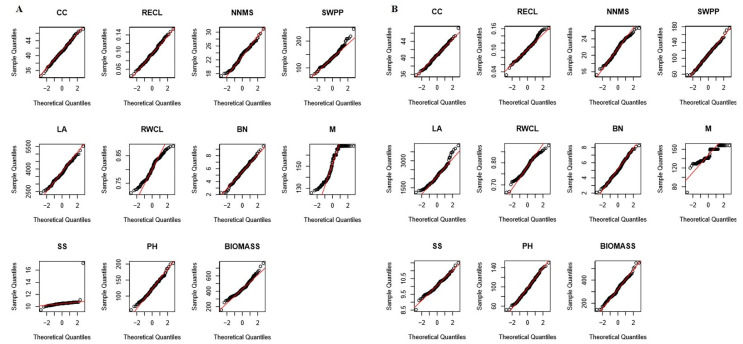
The QQ chart of normal distribution of 11 traits in 162 RILs under well-watered and drought stress conditions. (**A**) Distribution of traits under well-watered condition. (**B**) Distribution of traits under drought stress. LA, leaf area; PH, plant height; BN, branch number; CC, chlorophyll content; RWCL, relative water content of leaves; RECL, relative electric conductivity of leaves; NNMS, node number of main stem; BIOMASS, biomass; SWPP, seed weight per plant; PS, pull stem; M, maturity. The intercept of the red line is the mean and the slope is the standard deviation.

**Figure 2 ijms-23-10828-f002:**
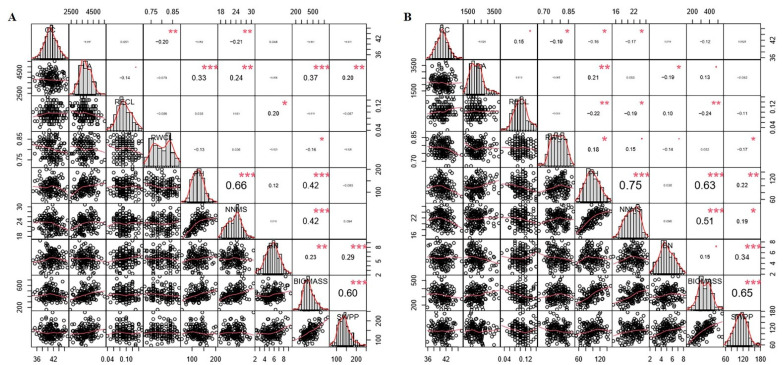
Pearson’s correlation coefficients (r) describing associations of two physiological traits and seven agronomic traits of 162 soybean genotypes evaluated under well-watered condition and drought stress condition. (**A**) The correlation analysis among the traits under well-watered condition. (**B**) The correlation analysis among the traits under drought stress condition. LA, leaf area; PH, plant height; BN, branch number; CC, chlorophyll content; RWCL, relative water content of leaves; RECL, relative electric conductivity of leaves; NNMS, node number of main stem; BIOMASS, biomass; SWPP, seed weight per plant. The diagonal line shows the distribution of the nine traits. The bivariate scatter plot with fitted lines is displayed below the diagonal line. The correlation coefficient and significant difference are shown above the diagonal line, and the higher correlation coefficient is, the greater the numerical code. The red dot * represents significant difference at *p* < 0.05; the red dot ** represents significant difference at *p* < 0.01; the red dot *** represents significant difference at *p* < 0.001.

**Figure 3 ijms-23-10828-f003:**
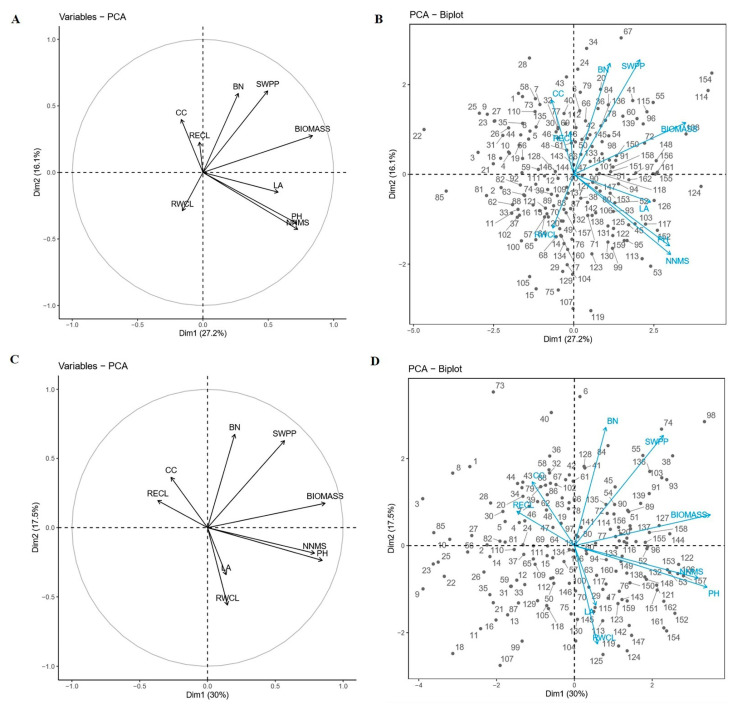
(**A**) The variables factor map of the nine traits under well-watered condition. (**B**) The principal component biplot displaying genotypic grouping under well-watered condition. (**C**) The variables factor map of the nine traits under drought stress condition. (**D**) The principal component biplot displaying genotypic grouping under drought-stress condition. LA, leaf area; PH, plant height; BN, branch number; CC, chlorophyll content; RWCL, relative water content of leaves; RECL, relative electric conductivity of leaves; NNMS, node number of main stem; BIOMASS, biomass; SWPP, seed weight per plant. The diagonal line shows the distribution of nine traits.

**Figure 4 ijms-23-10828-f004:**
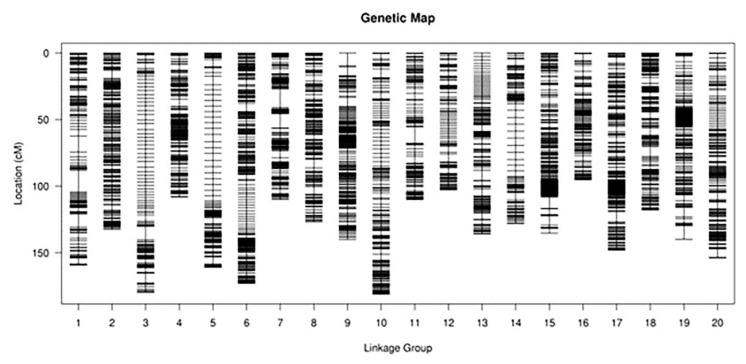
High-density linkage mapping of soybean RIL populations constructed by bin markers.

**Figure 5 ijms-23-10828-f005:**
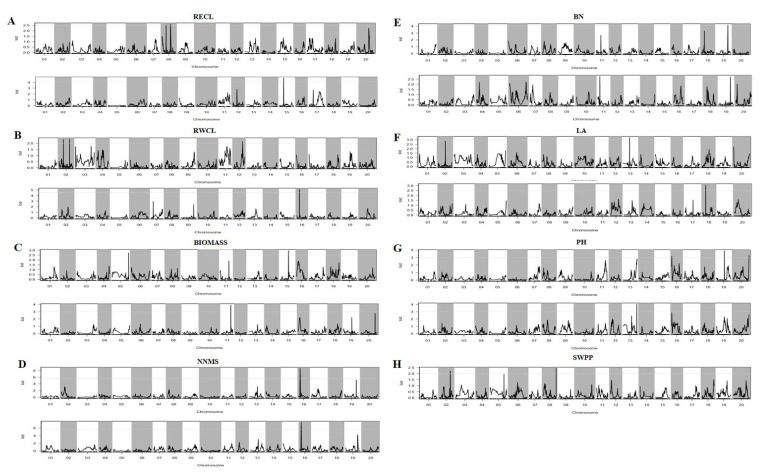
(**A**) RECL map positions (Mb) under well-watered condition and RECL map positions (Mb) under drought stress. (**B**) RWCL map positions (Mb) under well-watered condition and RWCL map positions (Mb) under drought stress. (**C**) BIOMASS map positions (Mb) under well-watered condition and BIOMASS map positions (Mb) under drought stress. (**D**) NNMS map positions (Mb) under well-watered condition and NNMS map positions (Mb) under drought stress. (**E**) BN map positions (Mb) under well-watered condition and BN map positions (Mb) under drought stress. (**F**) LA map positions (Mb) under well-watered condition and LA map positions (Mb) under drought stress. (**G**) PH map positions (Mb) under well-watered condition and PH map positions (Mb) under drought stress. (**H**) SWPP map positions (Mb) under drought stress.

**Figure 6 ijms-23-10828-f006:**
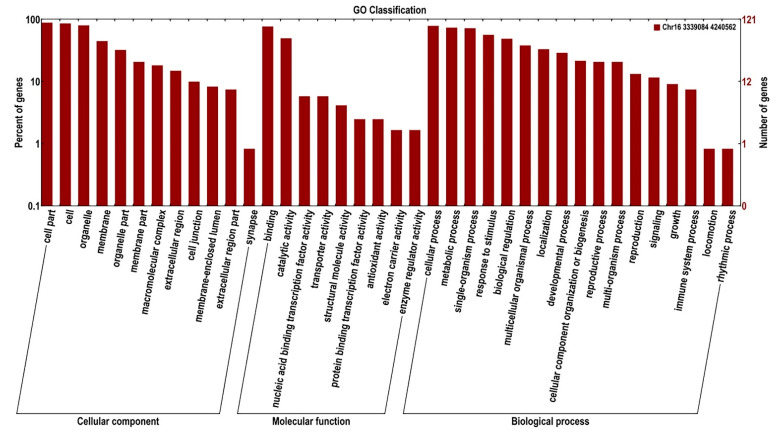
GO enrichment and classification of the candidate genes for NNMS under well-watered and drought conditions.

**Table 1 ijms-23-10828-t001:** Mean squares and significant tests after combined analysis of variance for nine agronomic traits under two water regimes.

Sources of Variation	DF		Seven Agronomic Traits						Two Physiological Traits	
		CC	LA	RECL	RWCL	PH	NNMS	BN	BIOMASS	SWPP
genotypes	161	22.707 **	1,158,244.424 **	0.002 **	0.006 **	3,295.922 **	33.930 **	8.982 **	37,381.694 **	2,779.981 **
Water status	1	61.573 **	622,894,650.534 **	0.013 **	0.096 **	171,580.293 **	669.760 **	55.676 **	3,034,425.977 **	228,060.573 **
genotypes × Water status	161	6.952 **	719,457.041 **	0.001	0.004 **	297.604 **	4.341 **	2.171 **	9,293.512 **	1,442.354 *

LA, leaf area; PH, plant height; BN, branch number; CC, chlorophyll content; RWCL, relative water content of leaves; RECL, relative electric conductivity of leaves; NNMS, node number of main stem; BIOMASS, biomass; SWPP, seed weight per plant. * represents significant difference at *p* < 0.05; ** represents significant difference at *p* < 0.01.

**Table 2 ijms-23-10828-t002:** Location and description of QTLs in RILs population derived from Jindou21 × Zhongdou33 grown under well-watered control (C) and drought stress (D) conditions.

Trait	Chr.	Start (cM)	End (cM)	LOD	Additive Effect	PVE (%)
LA-W	13	62.264	62.578	3.248	−143.48	5.436
LA-D	18	27.971	28.600	3.086	129.971	6.268
RECL-W	08	79.560	79.875	2.633	−0.005	5.874
RECL-D	15	39.392	39.706	4.744	0.007	9.220
RWCL-W	12	85.618	85.618	2.163	−0.008	3.757
RWCL-D	16	24.303	24.618	5.123	−0.014	10.063
PH-W	16	0.633	0.633	3.196	−8.135	8.187
PH-W	20	153.457	153.771	3.294	−8.326	8.577
PH-D	13	132.978	133.293	4.035	5.891	8.136
NNMS-W	16	13.769	13.769	8.681	−1.115	17.177
NNMS-D	16	13.769	13.769	7.510	−0.940	15.239
BN-W	19	113.603	113.603	4.209	0.386	7.454
BN-D	11	19.855	20.170	2.767	−0.307	5.311
BIOMASS-W	05	159.672	159.986	2.732	5.008	6.578
BIOMASS-W	15	81.000	81.250	2.955	−6.294	10.390
BIOMASS-D	11	89.689	90.318	3.930	4.915	7.6490
SWPP-D	02	104.790	105.355	2.230	4.457	3.846
SWPP-D	08	126.804	126.804	2.440	5.118	5.073

PVE, percentage of phenotypic variance explained by each QTL. LOD, logarithm of odds. LA, leaf area; PH, plant height; BN, branch number; CC, chlorophyll content; RWCL, relative water content of leaves; RECL, relative electric conductivity of leaves; NNMS, node number of main stem; BIOMASS, biomass; SWPP, seed weight per plant. The diagonal line shows the distribution of the nine traits; cM, centimorgan.

**Table 3 ijms-23-10828-t003:** Annotation and classification of candidate genes within NNMS-localized QTL intervals.

	Gene	Annotation	Annotation Homologous Genes	Function Annotation in Arabidopsis
			in Arabidopsis	
Kinase	*Glyma.16G044000*	mitogen-activated protein kinase kinase kinase kinase 3-like	unknown	MAP3
	*Glyma.16G040900*	adenylyl-sulfate kinase 3	*AT3G03900*	ASK3
Transport Factor	*Glyma.16G042900*	NAC domain-containing protein 100	*AT5G18270*	NAC100
	*Glyma.16G043200*	NAC domain-containing protein 18	*AT3G04070*	NAC18
	*Glyma.16G042300*	GATA transcription factor 9-like	*AT4G32890*	GATA9
	*Glyma.16G040000*	ethylene-responsive transcription factor RAP2-11-like	*AT5G18560*	RAP2-11
Function Protein	*Glyma.16G039900*	importin-5	*AT5G19820*	IPO5
	*Glyma.16G043700*	nucleoside-triphosphatase-like	*AT5G18280*	APY2
	*Glyma.16G043300*			
	*Glyma.16G043400*			
	*Glyma.16G043500*			
	*Glyma.16G039300*	lysosomal beta glucosidase-like	*AT5G04885*	unknown
	*Glyma.16G039400*			
	*Glyma.16G041200*	glutamate dehydrogenase 1	*AT5G18170*	GDH1
	*Glyma.16G043900*	transmembrane and coiled-coil domain-containing protein	*AT5G06660*	unknown
	*Glyma.16G036800*	BTB/POZ and MATH domain-containing protein 5	*AT5G21010*	BPM5
	*Glyma.16G036400*	BTB/POZ and MATH domain-containing protein 4	*AT3G03740*	BPM4
	*Glyma.16G036700*			
	*Glyma.16G038300*	methionine synthase	*AT5G17920*	ATMS1
	*Glyma.16G044400*	transportin-3	*AT1G12930*	TNPO3
	*Glyma.16G042000*	short-chain type dehydrogenase/reductase-like	*AT3G03980*	unknown
	*Glyma.16G037600*	2-aminoethanethiol dioxygenase	*AT5G39890*	unknown
	*Glyma.16G041900*	inhibitor of growth protein 4-like	*AT1G54390*	ING2
	*Glyma.16G036000*	cation/calcium exchanger 4-like	*AT5G17860*	CAX7
	*Glyma.16G036900*	50S ribosomal protein L25-like isoform 1	*AT4G23620*	RIPL
	*Glyma.16G040400*	arabinogalactan peptide 14-like	*AT5G56540*	AGP14
	*Glyma.16G040200*	5&apos-adenylyl sulfate reductase-like 5-like	*AT3G03860*	APRL5
	*Glyma.16G040900*	adenylyl-sulfate kinase 1, chloroplastic-like	*AT3G03900*	APK3
	*Glyma.16G040100*	tubulin alpha-3 chain-like	*AT5G19780*	TUA5
	*Glyma.16G041400*	putative H/ACA ribonucleoprotein complex subunit 1-like	*AT3G03920*	unknown
	*Glyma.16G039700*	cation/H(+) antiporter 15-like	*AT1G05580*	CHX23
	*Glyma.16G041300*	hypothetical protein PRUPE_ppa015420mg	*AT3G42170*	BED zinc finger
	*Glyma.16G036500*	hypothetical protein MTR_7g013520	unknown	unknown
	*Glyma.16G044300*	omega-hydroxy palmitate O-feruloyl transferase-like	*AT1G65450*	unknown
	*Glyma.16G039600*	quinate hydroxycinnamoyl transferase-like	*AT2G19070*	SHT
	*Glyma.16G038500*	mavicyanin-like	*AT3G17675*	unknown
	*Glyma.16G042500*	glucan endo-1,3-beta-glucosidase 8-like	unknown	unknown
	*Glyma.16G037200*	Phosphoribosyl glycinamide formyl transferase	unknown	unknown
	*Glyma.16G040800*	conserved hypothetical protein	unknown	unknown
	*Glyma.16G044200*	protein FLOWERING LOCUS T-like	*AT1G65480*	FT
	*Glyma.16G038400*	nucleolar MIF4G domain-containing protein 1-like	*AT5G17930*	unknown
	*Glyma.16G044600*	acetyl-coenzyme A synthetase-like	*AT5G36880*	ACS
	*Glyma.16G038200*	ankyrin-3-like	*AT2G31820*	unknown

## Data Availability

The datasets used and/or analyzed during the current study are available from the corresponding author on reasonable request. However, some of the data are shown in Appendix A.

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
