# Peer review of "Identification of Quantitative Trait Locus and Candidate Genes for Drought Tolerance in a Soybean Recombinant Inbred Line Population"

_ijms, 2022, doi:10.3390/ijms231810828_

Round 1
Reviewer 1 Report
In my opinion, the manuscript entitled “Identification of Quantitative Trait Locus and candidate genes for drought tolerance in a soybean Recombinant Inbred Line population”. falls within the journal scope. However, some shortcomings noted have been listed point wise to improve the manuscript.
Comments:
Homogenize the use of terminology throughout the manuscript
Add relations between physiological traits and drought in introduction section
Formatting from line 109-112 is not ok
Growth conditions missing in introduction sectio
Discussion is not well written it should be in accordance with results and abstract
Update introduction and discussion
Line 59 in introduction please use only botanical name.
Line 62, 63 add botanical name
References style is not uniform. For e.g Check journal name in case of references 1,2,3
Author Response
Responses to the Reviewer #1’s Comments
Q1
In my opinion, the manuscript entitled “Identification of Quantitative Trait Locus and candidate genes for drought tolerance in a soybean Recombinant Inbred Line population”. falls within the journal scope. However, some shortcomings noted have been listed point wise to improve the manuscript.
Response:
We are very grateful to you for your constructive comments and valuable suggestions that have greatly helped us improve the quality of our manuscript. We have carefully revised the manuscript accordingly to your comments and addressed them as you can see in the revised manuscript with green highlight and our point-to-point responses below.
Q2
Homogenize the use of terminology throughout the manuscript.
Response:
Thank you very much for your comments. The terminology has been revised throughout the manuscript.
Q3
Add relations between physiological traits and drought in introduction section.
Response:
Thank you very much for this constructive comment. Following your suggestion, we have added the relations between physiological traits and drought in introduction section of the revised manuscript as below:
“Drought tolerance is a complex quantitative trait. Many traits have changed under drought stress. Previous studies have indicated that some physiological traits such as relative water content, relative electrical conductivity and chlorophyll content and some yield-related traits such as plant height, number of nodes in main stems, leaf area, pod number, seed number and seed weight are affected by soil drought stress, in addition to the significant reduction in total dry matter and yield of soybean plants during the growth and flowering stages [70]. These traits can be considered as indicators to judge the drought tolerance of the crop [26, 71-72]. It is therefore necessary to have an evaluation of soybean phenotypes, physiological, and yield-related traits for drought tolerance.” (L.70-78)
Q4
Formatting from line 109-112 is not ok.
Response:
Thank you very much for your suggestion. The format of table 1 has been revised in the manuscript (L.128-132).
Q5
Growth conditions missing in introduction section.
Response:
Thank you very much for your suggestion. Plant growth conditions have been added to the last paragraph of the introduction as below:
“Some agronomic traits (node number of main stem (NNMS),chlorophyll content (CC), branches (BN), pull stem (PS), leaf area (LA), plant height (PH), biomass, seed weight per plant (SWPP) and maturity, and physiological traits (Relative water con-tent of leave (RWCL), Relative electric conductivity of leave (RECL) were analyzed for identifying QTLs and candidate genes for drought tolerance under well-water condition and drought stress condition, by using water-catch tank to simulate drought stress [57].” (L. 102-103)
Q6
Discussion is not well written; it should be in accordance with results and abstract. Update introduction and discussion.
Response:
Thank you very much for this constructive comment. The discussion has been revised according to your suggestion in the revised version as below:
“The use of genetic maps is essential for finding important loci, precision mapping and marker-assisted breeding [57]. Several genetic maps have been constructed for soybean based on molecular markers such as SSRs, ESTs, RFLP and RAPD markers [58-59]. Using DNA markers, it was possible to identify regions on the genetic map that could be identified by main genes, in accordance with standard mapping procedures. However, it is difficult to find the key locus associated with a specific trait by using non-specificity makes [60].
By advancing DNA sequencing technologies and applications, scientists have been able to improve plant breeding and aid in fine mapping processes in the past decade, as well as discover new types of molecular markers. Detecting SNP markers and accurate genotyping a high-density genetic map can be constructed with the next-generation sequencing technologies fast developed and the soybean reference genome sequence published [61-62]. The SNP molecular markers identification by WGRS has been well used for studying drought resistance in many important crops [30-31].
In this study, the parental lines and 162 RILs have been sequenced by WGRS to construct a high-density genetic map. Among them, the parental lines with a 20-fold sequencing depth and an average genome coverage of 90% or more, each RIL with 4.21-fold sequencing depth and a coverage of 92.86% or more. Ultimately, 32.84 Gbp of high-quality reads from Jindou 21, 33.3 Gbp of high-quality reads from Zhongdou33 and 748.99 Gbp of high-quality reads from their progeny were obtained. By utilizing bin markers and accurate genotypic data, it was possible to construct a high-density genetic map. An analysis of genotyping data showed that 4,843 recombination bin markers represented 923,420 SNPs on 20 linkage groups. There was an average distance of 0.57 cM between adjacent bin markers on the linkage map, with a total length of 2,737.51 cM. The collinearity of the genetic maps and physical maps was well for each linkage group (Supplementary Figure 2). Here, we demonstrate that the WGRS strategy is an effective tool for detecting markers and building high-density linkage maps. The WGRS mapping enabled us to obtain a great number of genome-wide SNPs, which accurately reflect the genetic diversity and genomic diversity of soybean. ” (L.243-272)
“In legumes, especially soybean, there are many studies on NNMS. NNMS is an important trait for soybean breeding. Soybean canopy and seed yield are determined by NNMS, which is one of the major plant agronomic traits [65]. Furthermore, there was a correlation between NNMS and other important agronomic traits, such as plant height, flowering, and maturity [66]. Li et al. reported that some QTLs associated with NNMS for plant density were identified using 144 four-way recombinant inbred lines (FW-RILs). The candidate genes were found on chr 06 and chr 19, named as Glyma.06G094400, Glyma.06G147600, Glyma.19 G160800.1, and Glyma.19G161100 [63]. Fu et al utilized 306 accessions from northeast China to identify 76 QTLs associated with NNMS for yield, and identified 49 candidate genes [64]. Plant height is one of the main hot spots in plant abiotic stress. There are many studies on the association between plant height and drought tolerance. In soybean, six QTL for drought tolerance associated with plant height have been identified (qPH2, qPH6, qPH7, qPH19-1, qPH19-2 and qPH19-3) [67-68].
In our study, there is a high relation with NNMS and PH (p<0.001) under well-watered condition drought stress condition. PH was also highly correlated with BIOMASS and Y traits under both conditions. The above result is consistent with earlier reports. Moreover, the correlation between NNMS and BIOMASS under drought stress condition was enhanced compared to well-water condition. It is tempting to speculate that NNMS and PH are traits associated with drought tolerance. ” (L.283-301)
Q7
Line 59 in introduction please use only botanical name.
Response:
Thank you very much for your suggestion. We have changed it as botanical name in the revised version in introduction.
Q8
Line 62, 63 add botanical name.
Response:
Thank you very much for your suggestion. We have added the botanical name in the revised version in introduction.
Q9
References style is not uniform. For e.g Check journal name in case of references 1,2,3
Response:
Thank you very much for this constructive comment. We have carefully checked all references to make them uniform in the revised version.
Reviewer 2 Report
This work will add depth to our understanding of QTLs that affect drought tolerance in soybean by using a recombinant inbred line (RIL) population from a cross between the drought-tolerant cultivar 'Jindou21' and - drought-sensitive cultivar 'Zhongdou33'. 9 agronomic and physiological traits were identified under drought and well-watered conditions. A total of five QTLs with logarithm of odds (LOD) value of 4.035-8.681 were identified on five chromosomes. Under well-watered and drought conditions, one QTL related to main stem node number was located on chromosome 16, accounting for 17.177% of the phenotypic variation. These results are very useful to understand the mechanism of drought tolerance in soybean. I suggest the manuscript should be accepted and published this journal.
Minor comments:
please check all of symbols, such as:
L152,L153 x
L333 " "
Figure 5 can not clearly present all of QTL information in genetic map.
More information on the part Construction of Linkage Map and QTL Analysis in methods, including as following:
how sequence?
how idenfiyt SNP?
how build genetic map, software or pipeline, and the parameters?
how identify QTL, software or pipeline, and the parameters?
Author Response
Responses to the Reviewer #2’s Comments
Q1
This work will add depth to our understanding of QTLs that affect drought tolerance in soybean by using a recombinant inbred line (RIL) population from a cross between the drought-tolerant cultivar 'Jindou21' and - drought-sensitive cultivar 'Zhongdou33'. 9 agronomic and physiological traits were identified under drought and well-watered conditions. A total of five QTLs with logarithm of odds (LOD) value of 4.035-8.681 were identified on five chromosomes. Under well-watered and drought conditions, one QTL related to main stem node number was located on chromosome 16, accounting for 17.177% of the phenotypic variation. These results are very useful to understand the mechanism of drought tolerance in soybean. I suggest the manuscript should be accepted and published this journal.
Response:
We are very grateful to you for your constructive comments and valuable suggestions that have greatly helped us improve the quality of our manuscript. We have carefully revised the manuscript accordingly to your comments and addressed them as you can see in the revised manuscript with green highlight and our point-to-point responses below.
Minor comments:
Q2
please check all of symbols, such as:
L152, L153 x
L333 " "
Response:Thank you for your careful review and comments to us, which have been helpful in promoting the content of our research. We have corrected characters through the manuscript.
Q3
Figure 5 can not clearly present all of QTL information in genetic map.
Response: Thank you very much for this constructive comment. Figure 5 has been revised to well present all of QTL information in genetic map. All QTL information has been placed on Figure 5 as follows.
Q4
More information on the part Construction of Linkage Map and QTL Analysis in methods, including as following:
how sequence?
how idenfiyt SNP?
how build genetic map, software or pipeline, and the parameters?
how identify QTL, software or pipeline, and the parameters?
Response: Thanks for your suggestions. The Information on the QTL map has been added in the Materials and Methods (L.393-447).
- how sequence? how identify SNP?
“DNA sequencing and SNP discovery
A total of 162 RILs, Jindou21 and Zhongdou33 genomic DNAs were extracted from fresh young leaves using the Plant DNA Kit (D2485, Omega). In order to sequence each DNA sample, paired-end sequencing libraries were constructed with an insertion of 300–500 bp. An Illumina Hiseq 2000 system (Illumina Inc., San Diego) was used to sequence the libraries with a 150 bp (PE150) read length. Low-quality reads (quality score < 20e) were filtered out and then raw reads were sorted to each sample according to barcode sequences. Using Samtools v1.9 [73] to mark duplicates, and GATK v 2.8.1 [74] to realign local elements and calibrate bases. With the default parameter settings, GATK and Samtools [73] SNP calling analysis was formed to produce a set of SNPs. There are eight segregation patterns based on polymorphic SNPs be-tween parents (ab×cd, ef×eg, hk×hk, lm×ll, nn×np, aa×bb, ab×cc, and cc×ab). Only SNPs with the aa×bb pattern were chosen for further analysis.” (L.393-405).
- how build genetic map, software or pipeline, and the parameters?
“Construction of Linkage Map
A total of 32.84 Gbp of clean data was obtained for parent Jindou21 and 33.3 Gbp for Zhongdou33. 160 offspring had a total of 748.99 Gbp of data. Q30 in 162 RILs reached over 80%. A total of 1913252 SNPs were detected between the parents and were screened for redundancy.
To determine recombination breakpoints and construct a bin map of RILs, a slightly modified sliding window approach was employed [75]. The ratio of SNPs with ‘Jindou21’ and ‘Zhongdou33’ genotypes was calculated. According to Huang et al. [75], a physical bin map was constructed based on recombination breakpoint position. Briefly, when RILs did not have recombination breakpoints within a 100 kb interval, these regions were combined into one bin. When recombination breakpoints didn’t oc-cur within 100 kb intervals, the regions were combined into a single bin.
A genetic map with ultra-high density was constructed using bins as markers. Molecular markers were divided into linkage groups (LGs) based on their locations on the genome. To further confirm the robustness of markers in each LG, modified loga-rithms of odds (MLODs) were calculated between markers. Before ordering, markers with MLOD scores < 5 were filtered. For resolving genotyping errors within LGs, the HighMap [76] strategy was utilized. The SMOOTH error correction strategy is then applied based on parents' genotype contributions, and a k-nearest neighbor algorithm is used to impute missing genotypes. After applying the multipoint method of maxi-mum likelihood, skewed markers were added to the map. Kosambi mapping function [77] was used to estimate map distances.” (L.414-427).
- how identify QTL, software or pipeline, and the parameters?
“QTL Analysis
The effects of genotype and environment, as well as interaction effects between geno-type and environment, were estimated using analysis of variances (ANOVA). It was estimated that broad sense heritability is derived from the formula: H2 =σ2g/(σ2g+σ2ge/n+σ2e/nr), where n delegates the number of environments, r delegates the number of replications, σ2g delegates the estimated genetic variance, σ2ge delegates variance for genotype-environment interaction and σ2e delegates experimental error. R package R/qtl has been used to identify QTLs via composite interval mapping (CIM) [78]. LOD values was determined based on a 1000-permutation test. QTLs were called for LOD values of 2 and higher. A 1000-permutation test was used to determine LOD values. When LOD values exceeded 2, QTLs were identified.” (L.428-438).
Reviewer 3 Report
The manuscript is prepared on a current topic, but it needs to be supplemented, or explain especially in the material and methodology section, because otherwise the results and their discussion are more difficult to grasp, or understandable. Abstract - the designation of concrete genes is given, which, however, do not say bic in the abstract. It is much more appropriate to indicate their annotated function in the organism and leave these counter-data to the results. Results and discussion - due to comments on the methodology, the results are difficult to grasp, including their subsequent discussion. It is appropriate to unify the labeling of genes (italics) and proteins into standards. Chaos in markings increases confusion. At the same time, I would expect more references to the last 5 years in the discussion, so that the novelty in knowledge is obvious and clear.
Material and methodology - the authors report the measurements of the control and treated variants, but do not precisely quantify the conclusion of the control and treated variants. By default, the % of water capacity of the substrate corresponding to the defined stress is defined. Here, during the experiment, there is a large fluctuation of values ​​(Figure S1), which can fundamentally affect the phenotypes of individual RILs and thus fundamentally the results achieved and their informative value. It is clear from Figure S1 that the differences were not always significant. With the device, it is also advisable to indicate the manufacturer and not only the types of devices. There is no indication why certain characters were chosen, this should be clear from the Introduction, but it is not. Section 5.4 is very short - it is advisable to add the parameters for fulfillment and the possibility of using individual steps. E.g. I do not assume the same yield of nucleic acids. Section 5.5 should add references or parameters for the Bin and Map functions. Line 342 lists abbreviations. Are they really all of them? After the formal page, I recommend checking the English language to remove typos (uppercase and lowercase letters, number formats, etc.) by a native speaker or a commercial entity. In the References section, the titles of the journals are not presented uniformly. Supplementary - e.g. Figure S1 - description of segments is missing, etc. - must be completed so that all figures are self-explanatory.
I do not recommend the manuscript for publication. If the trial conditions and individual evaluations are not clearly specified, it will not be possible to publish the manuscript!
Author Response
Responses to the Reviewer #3’s Comments
Q1
The manuscript is prepared on a current topic, but it needs to be supplemented, or explain especially in the material and methodology section, because otherwise the results and their discussion are more difficult to grasp, or understandable. Abstract - the designation of concrete genes is given, which, however, do not say bic in the abstract. It is much more appropriate to indicate their annotated function in the organism and leave these counter-data to the results. Results and discussion - due to comments on the methodology, the results are difficult to grasp, including their subsequent discussion. It is appropriate to unify the labeling of genes (italics) and proteins into standards. Chaos in markings increases confusion. At the same time, I would expect more references to the last 5 years in the discussion, so that the novelty in knowledge is obvious and clear.
We are very grateful to you for your constructive comments and valuable suggestions that have greatly helped us improve the quality of our manuscript. We have carefully revised the manuscript accordingly to your comments and addressed them as you can see in the revised manuscript with green highlight and our point-to-point responses below.
Q2
The manuscript is prepared on a current topic, but it needs to be supplemented, or explain especially in the material and methodology section, because otherwise the results and their discussion are more difficult to grasp, or understandable.
Response:Thank you for your comments. The materials and methods section has been revised and supplemented. (L.376-447)
Q3
Abstract - the designation of concrete genes is given, which, however, do not say bic in the abstract. It is much more appropriate to indicate their annotated function in the organism and leave these counter-data to the results.
Response:Thank you for your careful review and comments to us. We have added annotations of the candidate genes in the abstract and corresponded to the results as follows.
“Under well-watered condition and drought stress condition, one QTL related to main stem node number was located on chromosome 16, accounting for 17.177% of the phenotypic variation. Nine candidate genes for drought resistance were screened from this QTL, namely Glyma.16G036700.1, Glyma.16G036400.1, Glyma.16G036600.1, Glyma.16G036800.1, Glyma.13G312700.1, Glyma.13G312800.1, Glyma.16G042900.1, Glyma.16G043200.1 and Glyma.15G100700.1. These genes were annotated as NAC transport factor, GATA transport factor, BTB/ POZ-MATH proteins. This result can be used for molecular marker-assisted selection and provide a reference for breeding for drought tolerance in soybean.” (L.29-33)
Q4
Results and discussion - due to comments on the methodology, the results are difficult to grasp, including their subsequent discussion. It is appropriate to unify the labeling of genes (italics) and proteins into standards. Chaos in markings increases confusion. At the same time, I would expect more references to the last 5 years in the discussion, so that the novelty in knowledge is obvious and clear.
Response: Thank you very much for this comment. We have revised the discussion, references and methods according to your suggestions. In addition, inappropriate italics have been corrected in the results.
Q5
Material and methodology - the authors report the measurements of the control and treated variants, but do not precisely quantify the conclusion of the control and treated variants. By default, the % of water capacity of the substrate corresponding to the defined stress is defined. Here, during the experiment, there is a large fluctuation of values (Figure S1), which can fundamentally affect the phenotypes of individual RILs and thus fundamentally the results achieved and their informative value. It is clear from Figure S1 that the differences were not always significant.With the device, it is also advisable to indicate the manufacturer and not only the types of devices. There is no indication why certain characters were chosen, this should be clear from the Introduction, but it is not.
Response: Thank you very much for this constructive comment. We have replaced the line chart with a bar chart. This better shows the difference in soil water content after the two treatments. Furthermore, we have updated the data for soil water content from week 2 to week 22. From the bar chart we can see that the soil water content after the water-catch tank treatment decreases significantly from the week 7 onwards compared to the normal treatment. In addition, the measured values are influenced by the weather and show fluctuations. When rain falls, the soil water content rises in both the normal and water-catch tank treatment, however, soil water content in the water-catch tank treatment is consistently lower than normally treatment. The soil water content in the catchment tank treatment tended to stabilize and is much lower than in the normal treatment after a sustained drought. The manufacturer of the device has been added, and the introduction has been revised according to your suggestions. (L.102-103, L.376)
Supplementary Figure 1 Comparison of soil water content between normal and catchment tank treatments over 22 weeks. X-axis represents week of drought treatment, and y-axis represents soil water content (%).Statistically significant differences of soil water content between normal treatment and drought treatment (water-catch tank treatment) are marked with asterisks (*P<0.05, **P<0.01 and ***P< 0.001, Student’s t-test. ns, non-significant).
Q6
Section 5.4 is very short - it is advisable to add the parameters for fulfillment and the possibility of using individual steps. E.g. I do not assume the same yield of nucleic acids.
Response: Thank you very much for this constructive comment. We have made corrections and added contents to 5.4 as follows.
“5.4. DNA extraction, DNA sequencing and SNP discovery
A total of 162 RILs and Jindou21 and Zhongdou33 genomic DNAs were extracted from fresh young leaves using the Plant DNA Kit (D2485, Omega). In order to se-quence each DNA sample, paired-end sequencing libraries were constructed with an insertion of 300–500 bp. An Illumina Hiseq 2000 system (Illumina Inc., San Diego) was used to sequence the libraries with a 150 bp (PE150) read length. Low-quality reads (quality score < 20e) were filtered out and then raw reads were sorted to each sample according to barcode sequences. Using Samtools v1.9 [73] to mark duplicates, and GATK v 2.8.1 [74] to realign local elements and calibrate bases. With the default pa-rameter settings, GATK and Samtools [73] SNP calling analysis was formed to produce a set of SNPs. There are eight segregation patterns based on polymorphic SNPs be-tween parents (ab×cd, ef×eg, hk×hk, lm×ll, nn×np, aa×bb, ab×cc, and cc×ab). Only SNPs with the aa×bb pattern were chosen for further analysis.”
Q7
Section 5.5 should add references or parameters for the Bin and Map functions.
Response: Thank you very much for this constructive comment. We have added a method for constructing bin maps as follows.
“To determine recombination breakpoints and construct a bin map of RILs, a slightly modified sliding window approach was employed [75]. The ratio of SNPs with ‘Jindou21’ and ‘Zhongdou33’ genotypes was calculated. According to Huang et al. [75], a physical bin map was constructed based on recombination breakpoint position. Briefly, when RILs did not have recombination breakpoints within a 100 kb interval, these regions were combined into one bin. When recombination breakpoints didn’t oc-cur within 100 kb intervals, the regions were combined into a single bin.
A genetic map with ultra-high density was constructed using bins as markers. Molecular markers were divided into linkage groups (LGs) based on their locations on the genome. To further confirm the robustness of markers in each LG, modified loga-rithms of odds (MLODs) were calculated between markers. Before ordering, markers with MLOD scores < 5 were filtered. For resolving genotyping errors within LGs, the HighMap [76] strategy was utilized. The SMOOTH error correction strategy is then applied based on parents' genotype contributions, and a k-nearest neighbor algorithm is used to impute missing genotypes. After applying the multipoint method of maxi-mum likelihood, skewed markers were added to the map. Kosambi mapping function [77] was used to estimate map distances.”
Q8
Line 342 lists abbreviations. Are they really all of them?
Response: Thank you very much for this comment. We have added all the abbreviations that appear in the manuscript to the table as follows.
|
Abbreviations |
Full name |
|
RIL |
recombinant inbred line |
|
QTL |
quantitative trait locus |
|
SNP |
single nucleotide polymorphism |
|
LOD |
logarithm of odds |
|
WGRS |
whole genome resequencing techniques |
|
GWAS |
genome-wide association analysis |
|
Mb |
map positions |
|
LA |
leaf area |
|
PH |
plant height |
|
BN |
branch number |
|
CC |
chlorophyll content |
|
RWCL |
relative water content of leaves |
|
RECL |
relative electric conductivity of leaves |
|
NNMS |
node number of main stem |
|
BIOMASS |
biomass |
|
SWPP |
seed weight per plant |
|
PS |
pull stem |
|
M |
maturity |
|
GO |
Gene Ontology |
|
MAPK |
mitogen-activated protein kinase |
|
BPM |
BTB/POZ math |
|
FT |
flowering locus t |
|
ROS |
reactive oxygen species |
|
ASK |
adenylyl-sulfate kinase |
|
PEG |
polyethylene glycol |
|
MSR |
methionine sulfoxide reductase |
|
PBGD |
porphobilinogen deaminase |
|
LGs |
linkage groups |
|
MLODs |
modified logarithms of odds |
|
ANOVA |
analysis of variances |
|
CIM |
composite interval mapping |
Q9
After the formal page, I recommend checking the English language to remove typos (uppercase and lowercase letters, number formats, etc.) by a native speaker or a commercial entity.
Response: The English language has been checked and revised in the manuscript.
Q10
In the References section, the titles of the journals are not presented uniformly. Supplementary - e.g. Figure S1 - description of segments is missing, etc. - must be completed so that all figures are self-explanatory.
Response: Thank you very much for this comment. The titles of the references have been presented uniformly. We have replaced the line chart with a bar chart. This better shows the difference in soil water content after the two treatments. Furthermore, we have updated the data for soil water content from week 2 to week 22 as follows.
Supplementary Figure 1 Comparison of soil water content between normal and catchment tank treatments over 22 weeks. X-axis represents week of drought treatment, and y-axis represents soil water content (%).Statistically significant differences of soil water content between normal treatment and drought treatment (water-catch tank treatment) are marked with asterisks (*P<0.05, **P<0.01 and ***P<0.001, Student’s t-test. ns, non-significant).

Round 2
Reviewer 1 Report
the manuscript is suitable for publication.
reference 27 journal ist letter should be capital
Author Response
Responses to the Reviewer #1’s Comments
- reference 27 journal list letter should be capital
Response:
Thank you for your careful comments to us. We have corrected the format and serial numbering of references throughout the manuscript. The titles of the cited journals have also been corrected using the“Track Changes” function in the manuscript.
.
Reviewer 3 Report
The authors accepted all my comments. I am just pointing out the wrong numbering of references, when reference numbers 70 to 72 appear on lines 76-77. It is necessary to indicate the correct numerical order and correct the order in the References section. After editing, I recommend the yearbook for these publications. It is currently undergoing minor revision.
Author Response
Responses to the Reviewer #3’s Comments
Q1. the wrong numbering of references, when reference numbers 70 to 72 appear on lines 76-77.
Response:Thank you for your careful comments to us. We have arranged the serial numbering of the references in the full manuscript in order and corrected the wrong numbering of references using the “Track Changes” function in the manuscript.
Q2. It is necessary to indicate the correct numerical order and correct the order in the References section.
Response: Thank you very much for this suggestion. We have corrected the format and serial numbering of references throughout the manuscript. The titles of the cited journals have also been corrected using the“Track Changes” function in the manuscript.
.